# Data Analytics and Machine Learning Models on COVID-19 Medical Reports Enhanced with XAI for Usability

**DOI:** 10.3390/diagnostics15151981

**Published:** 2025-08-07

**Authors:** Oliver Lohaj, Ján Paralič, Zuzana Paraličová, Daniela Javorská, Elena Zagorová

**Affiliations:** 1Department of Cybernetics and Artificial Intelligence, Faculty of Electrical Engineering and Informatics, Technical University of Kosice, Letná 9, 042 00 Košice, Slovakia; jan.paralic@tuke.sk (J.P.); elena.zagorova@student.tuke.sk (E.Z.); 2Department of Infectology and Travel Medicine, Faculty of Medicine, L. Pasteur University Hospital, Pavol Jozef Šafárik University, Rastislavova 43, 041 90 Košice, Slovakia; zuzana.paralicova@upjs.sk (Z.P.); daniela.javorska@student.upjs.sk (D.J.)

**Keywords:** COVID-19 risk factors, COVID-19 severity prediction, COVID-19 mortality, machine learning, model explainability, risk prediction

## Abstract

**Objective**—To identify effective data analytics and machine learning solutions that can help in the decision-making process in the medical domain and contribute to the understanding of COVID-19 disease. In this study, we analyze data from anonymized electronic medical records of 4711 patients with COVID-19 disease admitted to hospital in Atlanta. **Methods**—We used random forest, LightGBM, XGBoost, CatBoost, KNN, SVM, logistic regression, and MLP neural network models in this work. The models are evaluated depending on the type of prediction by relevant metrics, especially accuracy, F1-score, and ROC AUC score. Another aim of the work was to find out which factors most affected severity and mortality risk among the patients. To identify the important features, different statistical methods were used, as well as LASSO regression, and explainable artificial intelligence (XAI) method SHAP values for model explainability. The best models were implemented in a web application and tested by medical experts. The model for prediction of mortality risk was tested on a validation cohort of 45 patients from the Department of Infectiology and Travel Medicine, L. Pasteur University Hospital in Košice (UNLP). **Results**—Our study shows that the best model for predicting COVID-19 disease severity was the LightGBM model with accuracy of 88.4% using all features and 89.5% using the eight most important features. The best model for predicting mortality risk was also the LightGBM model, which achieved a ROC AUC score of 83.7% and a classification accuracy of 81.2% using all features. Using a simplified model trained on the 15 most important features, the ROC AUC score was 83.6% and the classification accuracy was 80.5%. We deployed the simplified models for predicting COVID-19 disease severity and for predicting the risk of COVID-19-related death in a web-based application and tested them with medical experts. This test resulted in a ROC AUC score of 83.6% and an overall prediction accuracy of 73.3%.

## 1. Introduction

The COVID-19 pandemic has significantly impacted nearly every aspect of human life across the globe, presenting profound challenges not only in the medical field but also in social, technological, and economic domains. The causative agent, the SARS-CoV-2 virus, was first identified in December 2019 in patients with severe respiratory disease in Wuhan, Hubei Province, China [1]. By 11 March 2020, the World Health Organization (WHO) officially declared COVID-19 a global pandemic, prompting international public health responses and the recommendation to isolate individuals with respiratory symptoms, even in countries that had not yet reported any confirmed cases [2].

Since then, the pandemic has placed immense pressure on healthcare systems worldwide and highlighted the urgent need for effective tools to assist in medical decision making and patient triage. According to the WHO epidemiological update from 15 March 2024 [3], more than 774 million COVID-19 cases and over 7 million related deaths have been recorded globally, underlining the critical importance of continued research and preparedness.

In this work, we aim to explore the potential of machine learning and data analytics in predicting the severity of COVID-19 and the associated mortality risk based on electronic medical records. Our goal is to identify which models are most effective in supporting clinical decision making and to determine the most influential clinical and demographic factors contributing to severe outcomes in patients.

The structure of the article is as follows: the section Related Work provides a brief review of existing studies in this domain. The following section, Data Understanding, describes the dataset used, including its key features and characteristics. In Data Preprocessing, we detail the steps taken for cleaning, transforming, and selecting data. The section Data Modeling and Identifying Important Features is divided into two subsections: Modeling COVID-19 Severity and Modeling COVID-19 Mortality Risk. Each subsection outlines the models used, their performance metrics, and the methods applied for feature importance analysis. Simplified models using the most relevant features are also proposed to enhance practical applicability. Finally, the section Model Deployment and Testing on a New Cohort explains how the models were implemented in a web application and evaluated by medical experts using real-world patient data.

The novelty of our work lies in the combination of broad model comparison, explainability-focused modeling, and practical deployment in a clinical usability context. While earlier studies often limit themselves to a narrow algorithmic focus or lack post-model validation, we go further by (1) evaluating a wide spectrum of ML models, (2) simplifying them based on SHAP and domain expertise to ensure clinical applicability, (3) integrating explainable AI into a web-based tool, and (4) validating the system with real-world patient data and expert feedback. This end-to-end approach ensures not only performance but also usability, transparency, and real-world readiness. While expert consultation played a valuable role in validating feature relevance and interpretability, it was used in conjunction with statistical methods such as chi-square tests, ANOVA, LASSO regression, and SHAP values. As highlighted in Related Work, expert opinion alone is considered among the lowest levels of clinical evidence; robust machine learning in healthcare should integrate statistical modeling with empirical validation [4].

## 2. Related Work

Multiple studies have explored the use of machine learning (ML) on COVID-19 medical record data. Yu et al. [5] employed logistic regression and XGBoost models to predict mortality risk and assess the added value of respiratory support data. Similarly, Jha et al. [6] used XGBoost to forecast pulmonary fibrosis in COVID-19 patients. Sottile et al. [7] implemented a stacked generalization framework to improve mortality prediction over the SOFA score, using logistic regression with ridge regularization as the top-level model and several established clinical scores (SOFA, qSOFA, CURB-65, CCI, etc.) as component inputs.

Jiang et al. [8] investigated long COVID-19 risk prediction using vital sign time series from the first 7 days of hospitalization, applying PCA, Kolmogorov–Smirnov tests, XGBoost, and deep learning. Rivera et al. [9] explored the impact of statin use on COVID-19 severity using a super learner ensemble that included logistic regression, LASSO, random forest, and gradient-boosted trees.

In another study, Gao et al. [10] used LASSO regression for feature selection followed by Cox analysis to predict mortality at hospital admission, reducing the dataset to 34 key features. Butler et al. [11] employed SHAP values to identify key predictors of infection, ARDS, ICU admission, and mortality from structured EHR data. Zucco et al. [12] extracted over 2700 features from clinical records and selected the 22 most influential ones (e.g., age, BMI, number of diagnoses) using mean absolute SHAP values. A separate study [13] also emphasized the importance of using explainable AI (XAI) when working with sensitive health data.

Collectively, these studies underscore the role of ML in predicting COVID-19 outcomes such as mortality, complications, and long COVID, using techniques like XGBoost, random forest, and SHAP. However, many rely on a limited model set or focus on narrow outcomes. In contrast, our work employs a broader set of models—including LightGBM, CatBoost, KNN, SVM, logistic regression, and MLP—benchmarked using robust metrics (accuracy, F1-score, ROC AUC). We also integrate statistical techniques (e.g., LASSO) and XAI (SHAP) to enhance interpretability and identify risk factors. Most notably, we extend beyond modeling to practical implementation, deploying our best models in a web application tested with real-world patient data and clinical expert feedback. This bridges the gap between prediction performance and clinical usability.

## 3. Data Understanding

The dataset that was selected for this research consists of anonymized data from 4711 COVID-19-positive patients from a hospital in Atlanta (available at: https://www.kaggle.com/datasets/harshwalia/mortality-risk-clinincal-data-of-covid19-patients/data, accessed on the 15 May 2024). The dataset was made publicly available by Harsh Walia on the Kaggle platform, where it is licensed for unrestricted use in academic and research applications. It consists of anonymized clinical records from hospitalized COVID-19 patients and has been cited in several ML studies. The original dataset contained 85 features, including data from medical reports, clinical laboratory measurements, data about comorbidities, and general patient data. After discussion with the domain expert, we found that the dataset also contained so-called supplementary features, which provide information on whether a given laboratory measurement was performed or not, and other supplementary features indicating whether a given laboratory measurement value of the patient exceeded the normal range. Table 1 lists all the features in the dataset excluding the supplementary features. For each feature there is description completed in collaboration with the expert, as well as the feature type.

Binary features have a value of 0 or 1 indicating whether the feature applies to the patient or not. mg = milligram, mmHg = millimeter of mercury, mL = milliliter, dL = deciliter.

The age of the patients ranged from 18 to 103 years, and the mean age was 64 years. Figure 1 shows the age distribution of the patients, color-coded into 4 age groups: under 60 years, 61 to 70 years, 71 to 80 years, and over 80 years.

### 3.1. Data Cleaning

In the original dataset, the missing values were marked with a 0, so it was necessary to first determine in which cases the number 0 is indeed a value of 0 and where it is a missing value. After research, it was determined that all values of the laboratory measurements given as 0 can be considered as missing values. There were also several instances of incorrect values in the original dataset, the most obvious of which was the body temperature listed several times with the number −17.777. Similarly, for the INR, the international normalized ratio, several values were 17.0001, which is also a very unlikely value. In such cases, it was probably an incorrect entry, and these values were also considered as missing. After identifying all the missing values, we analyzed their percentage of occurrence in each feature. Overall, 4.37% of values were missing in the dataset. Features with more than 30% missing values—such as Procalcitonin and Interleukin-6—were excluded to reduce imputation bias. This follows best practices in clinical machine learning, where sparsely populated variables can degrade model performance and introduce instability during training [14,15].

To maintain data integrity and avoid selection bias, we retained all patient records and imputed missing values using multiple imputation by chained equations (MICE). This approach is recommended for electronic health record datasets in the context of COVID-19, where patient-level deletion may reduce statistical power and introduce bias [16]. Therefore, we used the MICE approach, which is widely adopted for handling missing values in clinical datasets, including COVID-19-related studies [17].

To prevent information leakage during imputation, we ensured that the target variable (“severity” or “death”) was excluded from the imputation process.

### 3.2. Data Transformation and Selection

For statistical analyses where the assumption of normality was required, all continuous numerical features were transformed to approximate a normal distribution using a log transformation. This step was essential for ensuring the validity of parametric tests, which rely on the assumption that data are normally distributed. The log transformation helps reduce skewness, stabilize variance, and improve the symmetry of the data distribution, thereby making the statistical tests more robust and reliable.

In preparation for machine learning modeling, the numerical input features from the original dataset were normalized to a standardized range between 0 and 1 using min–max scaling. This normalization ensures that all features contribute equally to the model and prevents features with larger scales from dominating the learning process. This is particularly important for distance-based algorithms and gradient descent optimization methods, which are sensitive to the scale of input variables.

Categorical variables were also preprocessed to ensure compatibility with machine learning algorithms. Binary categorical variables were encoded as 0 and 1. Categorical features with more than two distinct classes were transformed using one-hot encoding, resulting in the creation of separate binary indicator features for each category. This transformation allows the model to interpret and learn from categorical data without imposing an arbitrary ordinal relationship among the classes.

## 4. Modeling and Identifying Important Features

### 4.1. Modeling COVID-19 Severity

COVID-19 severity in the original dataset was on a scale of 0 to 11 (Figure 2), with a mean of 3.6. The *x*-axis represents the COVID-19 severity, and the *y*-axis represents the number of patients.

#### 4.1.1. Chi-Square

A chi-square test was used to find if there was a statistically significant influence of the categorical independent features on severity. Table 2 shows all categorical features whose *p*-value was less than 0.05, i.e., they had a statistically significant effect on the severity of COVID-19 disease among patients in dataset. The most significant features were age category (Age_category), and creatinine score (CrtnScore) whose *p*-values were very close to 0.

#### 4.1.2. ANOVA

An ANOVA test was performed to find if there was a statistically significant influence of the numeric independent features on severity. Table 3 shows all numerical features whose *p*-value was less than 0.05. The most significant features were age (Age_number) and blood urea nitrogen (Blood_urea_nitr) whose *p*-values were very close to 0.

This three-level stratification (low: 0–2, moderate: 3–5, high: 6–11) aligns with typical clinical severity groupings used in COVID-19 case management guidelines, particularly those published by the World Health Organization [18]. As can be seen in the graph (Figure 3), class 0 includes the severity interval 0 to 2 from the original dataset, class 1 includes severity 3 to 5, and the highest class 2 includes severity 6 to 11.

#### 4.1.3. All Models’ Performance

We used the following models to predict COVID-19 disease severity: random forest, LightGBM, XGBoost, KNN, support vector classifier, logistic regression, and MLP neural network. All models were built on a training and testing set split in a ratio of 80:20. In the following Table 4, we report only the accuracy metric, as it was used to establish a baseline comparison across all models. Based on this initial comparison, we identified the model with the highest accuracy and conducted a more detailed analysis of its performance, as seen in the following subsection.

#### 4.1.4. Best Model for Severity Prediction

The best model was LightGBM, with an accuracy on the test set of 0.884. Table 5 shows the results of each metric when testing this model. The first three rows show the different target feature classes: low, moderate, and high severity, followed by the average of the metrics for all classes (macro_avg) and the weighted average (weighted_avg), which considers the number of examples in each target class. The three columns contain the metrics for measuring model performance, and the last column lists the number of test examples for each class or the total number of examples for the average of the metrics in the last two rows.

As seen in Table 5, the model had the highest precision in predicting low severity (class 0): a precision of 0.94, which means that 94% of the examples predicted as class 0 were actually class 0. For class 1 (moderate severity), precision was 0.88 and for class 2 (high severity) it was 0.81. In terms of recall, for class 0, 93% of the true class 0 examples were also correctly classified, followed by 85% for class 1 and 87% for high severity. The F1-score, i.e., the harmonic mean of the precision and recall metrics, was, for class 0, 0.93; for class 1, it was 0.87; and for class 2, it was 0.84.

#### 4.1.5. SHAP Values for Severity Modeling

Using SHAP values for the LightGBM model, we investigated the importance of features for the prediction. Consistent with current guidance, we did not apply an arbitrary SHAP value cutoff; instead, we prioritized interpretability and clinical consensus, aligning with best practices for explainable AI in healthcare [19]. As the target feature consists of three classes, for simplicity we show a plot of the influence of each feature on only target class 2, the highest COVID-19 severity. The graph (Figure 4) shows the most influential features, with blue representing low values for a given feature and red representing high values. The *x*-axis shows the SHAP values for each feature. Negative SHAP values (to the left of the vertical axis at point 0) indicate that the given feature values “push” the prediction towards a lower prediction of the target feature—i.e., in our case, a lower severity. Conversely, positive SHAP values (to the right of the vertical axis at point 0) indicate that the given feature values are “pushing” the prediction towards a higher severity. For example, in the case of the feature age (Age_number), we can see that a lower age of patients (purple-blue color) tended towards a lower severity of COVID-19 and a higher age of patients tended towards a higher severity of COVID-19. The opposite is the case for the feature oxygen saturation (Oxygen_sat). The graph clearly shows that high saturation values (red) tended to lower COVID-19 disease severity and lower oxygen saturation (purple and blue) in turn tended to increase severity.

#### 4.1.6. Simplified Model for Severity Modeling

In order to deploy the model in the application and increase usability, it was necessary to create a simplified model that was trained only on the most important features for severity detection. After identifying the most important features for COVID-19 disease severity prediction using the LightGBM model and SHAP values and consulting with a domain expert, te following 8 features were selected: age (Age_number), oxygen saturation (Oxygen_sat), C-reactive protein (CRP), blood aspartate aminotransferase (AST), D-dimer (Ddimer), international normalized ratio (Inter_norm_ratio), creatinine (Creatinine), and platelets (Platelets). We omitted features creatinine score (CrtnScore_0) and platelet score (PltsScore_0) features because it was not clear in the dataset description by which thresholds these scores were given and also the values for both of these measurements are already included in the selected important features.

Through an iterative process of hyperparameter tuning using 5-fold cross-validation, we found that the simplified LightGBM model performed best with the following parameters: learning_rate = 0.1, max_depth = −1, n_estimators = 350, num_leaves = 8.

The simplified model achieved an overall accuracy of 0.895 on the test set, which is higher than the model trained on a larger set of predictors. This result may be because, with a large number of predictors, noise may arise that causes misclassification of some examples in the dataset. Since the target *Severity_class* feature contains 3 classes, we used the accuracy, recall and F1-score metrics for each class to display the results.

Table 6 shows the model results for each target class. As we are able to see, opposed to the previous results mentioned in Table 5, the precision for class 2 has improved. When we look into recall, we can see that two classes, classes 0 and 1, have improved in results. In the F1-score metrics, all classes have seen improvement. Support represents the number of instances from the test set in each target class that were used to calculate these metrics.

Based on the data from the model performance metrics, it can be concluded that the simplified model best predicts class 0 with the highest precision, recall, and F1-score. Classes 1 and 2 have lower values in all metrics, indicating that they are more challenging for the model to predict.

### 4.2. Modeling COVID-19 Mortality

To predict mortality risk of patients, we used the binary target feature “death”, which contains the value 0 if the patient did not die and 1 if the patient died during hospitalization. From 4711 COVID-19-positive patients in the dataset, 1148 died and 3563 did not die.

#### 4.2.1. Chi-Square Test

The following Table 7 presents the chi-square test results for the effect of categorical features on patients’ mortality risk, showing only features with a *p*-value less than 0.05.

#### 4.2.2. *t*-Test

We used Student’s *t*-test to determine whether there was a statistically significant difference in the mean value of the features between patients who died versus those who did not. In Table 8, we report the results of Student’s *t*-test for features that have a *p*-value less than 0.5.

#### 4.2.3. LASSO Logistic Regression

Using LASSO regression, we investigated which features had the greatest impact on patients’ mortality. At the top of the graph (Figure 5) are the features with positive coefficients (increasing the mortality risk), and at the bottom are the features with negative coefficients (decreasing mortality risk).

#### 4.2.4. All Models for Mortality Risk Prediction

Based on the current state analysis and the conclusion of it, we have decided to use the following data analytics and machine learning models to predict the risk of death in COVID-19 patients: random forest, LightGBM, XGBoost, KNN, Naïve Bayes, SVC, logistic regression, and MLP. As this is a binary classification task where target class 1 is underrepresented, it was considerably more difficult for the models to find and correctly classify patients who belonged to this class. Therefore, in addition to the accuracy metric, we used the F1-score metric for class 1 and also the ROC AUC score to evaluate the models. In the following table, we can see the results of the baseline model’s performance estimated by cross-validation on the training set, ordered by F1-score of class 1:

As we can see in Table 9, the random forest model had the highest classification accuracy (precision) and ROC AUC scores, but the LightGBM model had a three percent higher F1 class 1 score, with very similar precision metrics and ROC AUC scores. Because the LightGBM model was better able to detect the risk of patient death than the random forest model, we consider the LightGBM model to be the better fit for this problem.

#### 4.2.5. The Best Model for Classification

The overall classification accuracy of the LightGBM model on the test set is 0.812, while the ROC AUC score on the test set was 0.837. Table 10 shows the values of other metrics for the LightGBM model on the test set.

In this binary classification task, recall for class 1 corresponds to sensitivity (true positive rate), and recall for class 0 corresponds to specificity (true negative rate). These values are reported in the table above. Table 10 also shows that the LightGBM model trained on all features achieved a precision of 0.83 in classifying examples of target class 0 and also a high recall of 0.93. There were 699 examples in this class from the test set. Regarding the classification of patients into class 1, the model achieved a reasonably good precision of 0.71, but a low recall of 0.46. This means that the model was able to correctly identify 93% of the patients in the test set who did not die and 46% of the patients who died. This may be because there were significantly fewer patients in the target class 1 in the dataset or because the original dataset contained relatively few comorbidities, lacking data on diseases such as obesity or cancer, which have been shown to have an impact on mortality in COVID-19 patients. To provide a clearer picture of model behavior in the context of class imbalance, we also report sensitivity and specificity. The full LightGBM model achieved a sensitivity of 46% and a specificity of 93% on the test set, highlighting its strong performance in identifying non-mortality cases, while still capturing nearly half of the actual mortality cases.

In Figure 6, a visualization of the area under the ROC curve can be seen. The curve is generated by plotting the true positive rate (cases that were in class 1 and correctly identified as being from class 1) against the false positive rate (cases that were in class 0 but classified as class 1 by the model) at different threshold settings.

#### 4.2.6. SHAP Values for Mortality Modeling

Figure 7 shows the most influential features according to SHAP values used for the LightGBM model. For features that are predominantly red to the right of the center axis, this means that a higher value of that feature contributed to a higher risk of death in patients. Conversely, if the values to the right of the center axis are blue, it means that higher values contributed to a lower risk of death. Feature mean arterial pressure (Mean_art_press) had the greatest effect on prediction, and with a lower pressure, the chance of death increased. This may be related to the fact that, as oxygen saturation decreases, blood pressure also decreases. Another important feature was age (Age_number), with the chance of death increasing with age. These findings are consistent with prior studies identifying the same clinical features—such as age, oxygen saturation, CRP, D-dimer, and kidney function markers—as significant predictors of COVID-19 severity and mortality [9,10,12].

#### 4.2.7. Simplified Model for Mortality Modeling

As the original model was trained using 51 predictors, it was necessary to reduce this number for deployment in the application to make it easier to enter new patient data and to increase the usability of the model in practice. The following 15 most important features based on the SHAP values and consultation with medical experts were selected for deploying the model to predict the risk of death in patients:Mean arterial pressure (Mean_art_press);Patient age (Age_number);C-reactive protein level in the blood (CRP);Number of days of hospitalization (Length_of_stay);Creatinine level (Creatinine);Oxygen saturation (Oxygen_sat);Blood urea nitrogen level (Blood_urea_nitr);Blood aspartate aminotransferase (AST);Body temperature (Temperature);Ferritin level (Ferritin);Glucose (Glucose);Troponin (Troponin);Alanine aminotransferase in the blood (ALT);Blood D-dimer level (Ddimer);Platelets (Platelets).

We excluded the feature peripheral vascular disease (Per_vasc_dis_0) because the simplified model will be fitted and tested on a group of patients for whom this was not directly reported in the documentation.

By tuning the hyperparameters, we found that the simplified model performed best in F1-scores at the following settings: reg_alpha = 2, reg_lambda = 1, min_data_in_leaf = 50. This model achieved an overall ROC AUC score on the test set of 0.836, which is almost identical to the model trained on all features (0.837). The overall classification accuracy of the simplified model was 0.805. Table 11 shows that the simplified model has increased the F1-score of the class 1 compared to the model trained on all features, from 0.56 to 0.60.

The simplified model achieved improved sensitivity of 57% while maintaining specificity of 89%. This increase in sensitivity is particularly important in clinical use, as it indicates better performance in identifying high-risk patients, even with fewer input features.

## 5. Model Deployment and Testing on New Cohort with Experts

### 5.1. Model Deployment

To deploy the two simplified models for severity and mortality risk prediction, we created a web application consisting of the following three parts:Machine learning models;Flask application;HTML and CSS for user interface.

The Flask application was implemented using Python’s Flask 2.2 library to serve the simplified LightGBM models through a lightweight API. In the application, we used two models that were evaluated and approved by a domain expert who said that the features used in the models and their impact on severity or risk of death correspond to medical practice. The application had two main sections, one for the severity prediction and the other for the mortality risk prediction. In each section, there is a description of the model and form for the patient data. After adding the patient data, a prediction result is shown and, for mortality risk prediction, the risk is displayed as percentual probability of a patient belonging to class 1. For each model there is also shown a SHAP values plot for individual prediction, which displays how features contributed to the prediction result of the patient. The severity prediction section can be seen in Figure 8 and Figure 9. The mortality prediction section is similar, with a different model used.

### 5.2. Model Testing with Experts

The deployed models in the app were tested in person with two experts from medical practice and also using anonymized data of 45 patients with COVID-19 from the Department of Infectiology and Travel Medicine, L. Pasteur University Hospital in Košice (UNLP). This study was approved by the Ethic Committee of the UNLP on 19 February 2024. We used these data to test a model to predict the risk of death. The aim of the testing was to determine whether the developed models were credible and potentially applicable in medical practice.

During testing with two clinical experts from L. Pasteur University Hospital in Košice, no major usability issues were reported. The experts found the interface intuitive and appreciated the integration of SHAP-based explanations for each prediction. They also expressed that the ability to input patient data directly and receive real-time model output enhanced the system’s potential for clinical use. Some suggestions were made for future improvements, including clearer labeling of some clinical features (e.g., using standardized medical terminology), the option to export predictions to patient records, and the display of confidence intervals or prediction certainty levels. These suggestions are valuable and will be considered in the next phase of system development.

The overall accuracy of the severity prediction model tested on a group of 45 patients from the UNLP could not be quantified because the patients were not divided into three severity groups in this hospital. The summary severity prediction for the 45 patients was shown to the experts, visually checked, and then the model was tested on several patient data individually using our application, which also displayed the factors influencing a given patient’s COVID-19 disease severity. Overall, the model was able to identify the COVID-19 disease severity of individual patients to a high degree of accuracy. The model was also able to identify improvement or worsening of the condition in some of the patients for whom repeated measurements were taken.

The overall ROC AUC score achieved by the model for predicting the risk of death in a group of 45 patients from UNLP was 0.836 (83.6%), and the overall accuracy was 0.733 (73.3%). The lower accuracy rate was probably due to the model failing to identify high risk of death if there were reasons not included in the model or in the original dataset (e.g., muscle bleeding or pulmonary embolism).

### 5.3. Results

Based on the testing, both experts expressed a positive impression of the models in the application, with higher confidence for the COVID-19 severity prediction model than for the prediction of mortality risk. This corresponds with the performance of the models, as the model for severity prediction already had better results during the modeling phase. As mentioned above, for a more accurate prediction of the risk of death, it is necessary to develop a more comprehensive model and to include as much additional data as possible on comorbidities or other risk factors of patients. Regarding potential use in practice, the experts also expressed that our models have potential for use in medical practice.

After the testing, possible use cases were discussed where such models could be used:In a crisis situation, such as the beginning of the COVID-19 pandemic, the models could assist in identifying the risk of patients’ conditions.To investigate changes in patients’ condition (continuous display of model results based on repeated patient measurements).To identify possible influential factors using similar models and explainability methods for other diseases.

## 6. Discussion

In this study, we applied machine learning (ML) and data analytics methods to structured clinical data from 4711 COVID-19 patients treated at a hospital in Atlanta. Our goal was to identify the most effective models for predicting disease severity and mortality and to determine the key clinical features influencing these outcomes.

The original dataset included 85 features, which were refined to 53 through feature selection and preprocessing steps such as MICE-based imputation and categorical encoding. Feature significance was assessed using statistical methods including chi-square tests, ANOVA, *t*-tests, and LASSO regression.

For severity prediction, the LightGBM model performed best, with an accuracy of 88.4%. A simplified version using the top eight features slightly improved accuracy to 89.5%, while enhancing interpretability and clinical usability through SHAP-based explainability.

For mortality risk, LightGBM again achieved the highest performance (ROC AUC 83.7%, accuracy 81.2%). The simplified model using 15 key features achieved nearly identical results (ROC AUC 83.6%, accuracy 80.5%) and showed improved F1-score for the minority class, indicating better handling of class imbalance.

When compared to recent studies in the field, such as those by Moulaei et al. [20], Sun et al. [21], and Baik et al. [22], which achieved AUROC values between 0.86 and 0.99 using advanced ensemble or deep learning methods, our LightGBM model performed slightly worse in terms of AUC. In a different approach, Kim et al. [23] built a mortality prediction model using logistic regression and machine learning techniques, with their linear SVM model reaching an AUC of 0.962 while using only 10 key features. However, our focus was on achieving a balance between predictive performance and interpretability, which is critical for real-world medical application. Unlike many black-box models, our approach offers transparency, ease of understanding, and expert validation—qualities valued in clinical settings.

Importantly, both simplified models were reviewed by clinical experts from L. Pasteur University Hospital (UNLP) in Košice, who confirmed their relevance and practical utility. To further validate the mortality model, we tested it on an external cohort of 45 anonymized patients from UNLP, yielding a ROC AUC of 83.6% and accuracy of 73.3%. These results confirm the robustness and generalizability of the model in real-world conditions.

Despite these promising results, our study has limitations. Our study includes external validation using anonymized patient data from a different hospital and real-world testing with clinical experts. While the size of the validation cohort (45 patients) is modest, it provides valuable insight into the practical usability of our models beyond the original dataset. This type of validation—expert-reviewed and contextually grounded—is an essential step toward real-world clinical deployment. Future work will expand this validation across additional patient populations to further support generalizability. Furthermore, some potentially influential variables—such as viral strain information, socioeconomic status, or vaccination status—were not included. Future research could expand the dataset, integrate multimodal data sources (e.g., imaging, genomics), and evaluate additional model architectures including transformer-based or hybrid deep learning frameworks. Benchmarking against high-performing published models using standardized datasets and protocols is also a planned next step.

## 7. Conclusions

This research demonstrates the effectiveness of machine learning and data analytics methods in predicting COVID-19 severity and mortality based on structured clinical data. LightGBM models proved to be particularly effective, and through feature selection and explainability techniques (such as SHAP values), we developed simplified models that retain high performance while enhancing interpretability and clinical usability.

Crucially, the models were validated through expert feedback and tested on external patient data, demonstrating real-world applicability. Our work highlights the importance of balancing predictive performance with model transparency—especially in high-stakes domains like healthcare—where usability by medical professionals is essential.

The findings support the potential of explainable ML models to assist clinicians in risk stratification and decision making. Future research should focus on broader validation, integration with clinical workflows, and expanding feature sets to include more diverse patient information. With continued development, such tools could significantly support healthcare systems in managing COVID-19 and similar diseases.

## Figures and Tables

**Figure 1 diagnostics-15-01981-f001:**
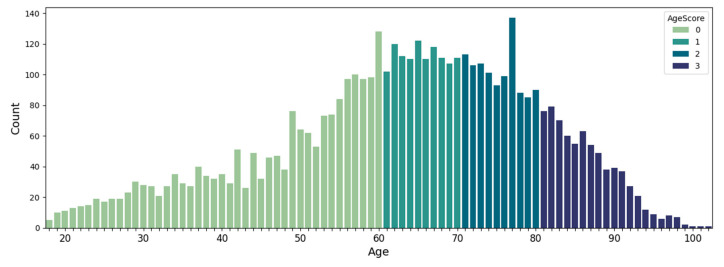
Distribution of patients’ ages in dataset.

**Figure 2 diagnostics-15-01981-f002:**
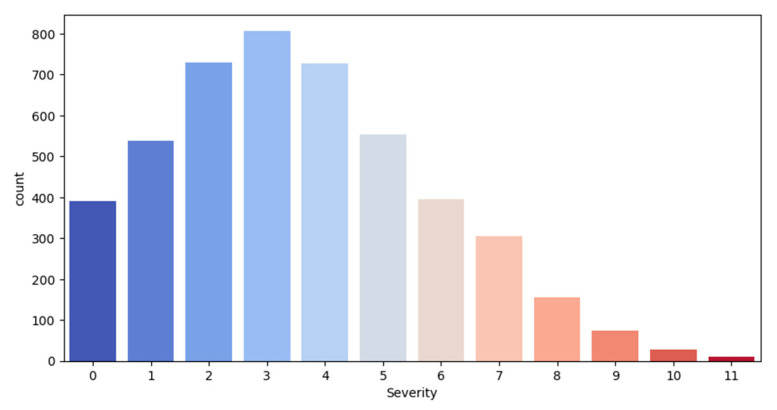
Severity distribution in the dataset, from lowest (0) to highest (11).

**Figure 3 diagnostics-15-01981-f003:**
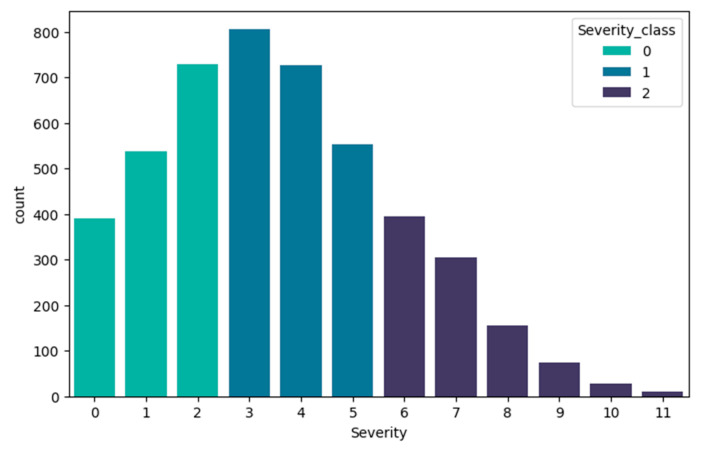
Target feature Severity class.

**Figure 4 diagnostics-15-01981-f004:**
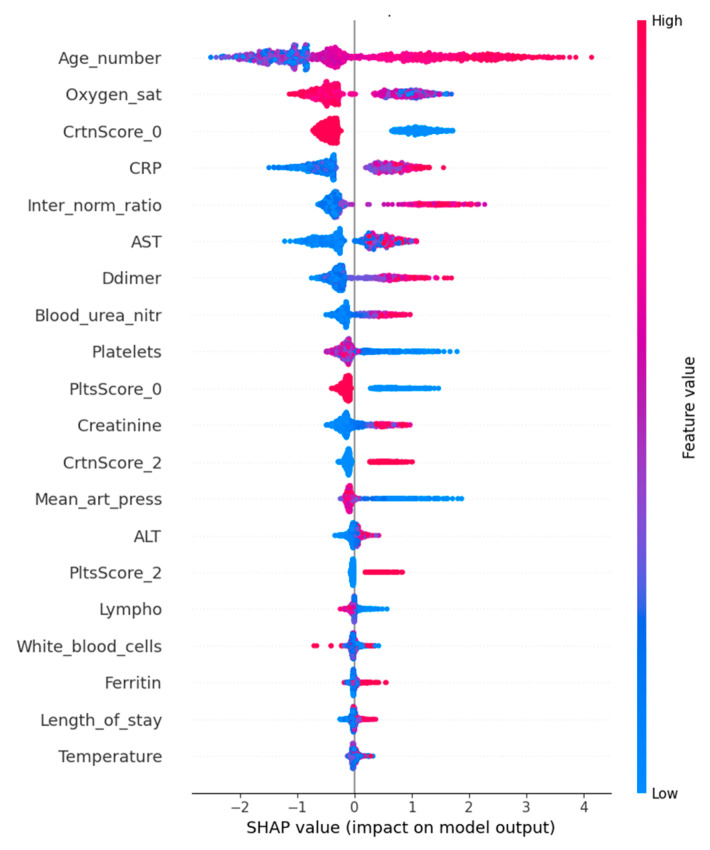
SHAP values for most important features for Severity prediction.

**Figure 5 diagnostics-15-01981-f005:**
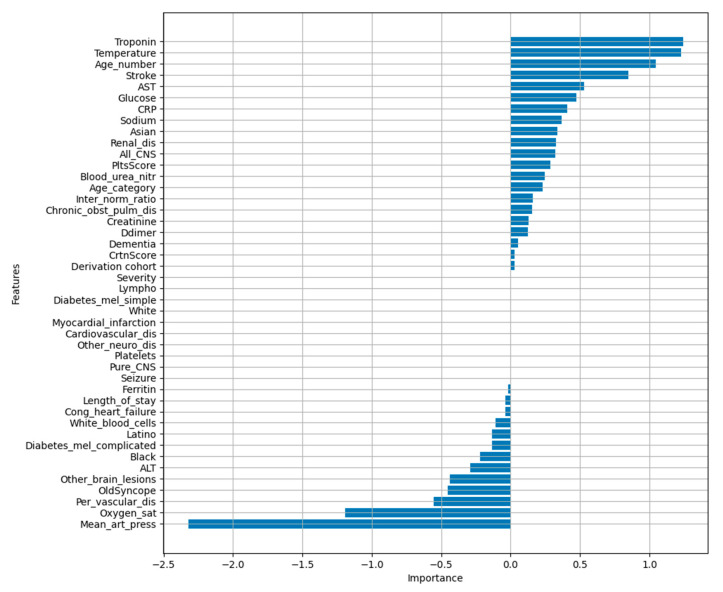
LASSO logistic regression results for Mortality risk.

**Figure 6 diagnostics-15-01981-f006:**
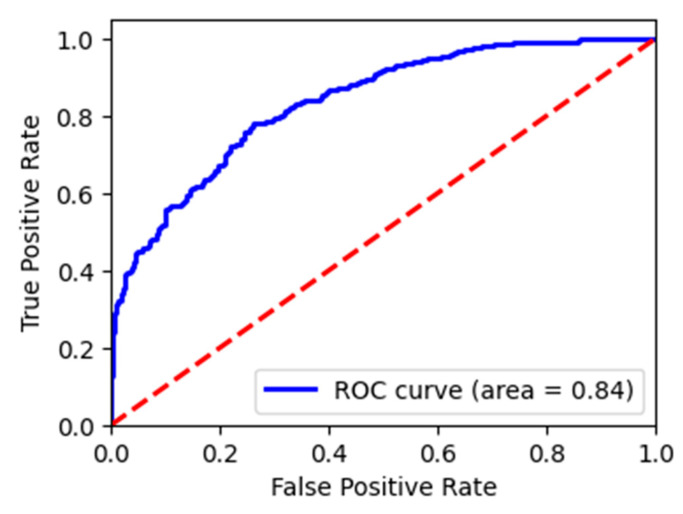
Area under the ROC curve of LightGBM model for mortality risk prediction. The red dotted line represents the line of no-discrimination (AUC = 0.5), indicating random classifier performance.

**Figure 7 diagnostics-15-01981-f007:**
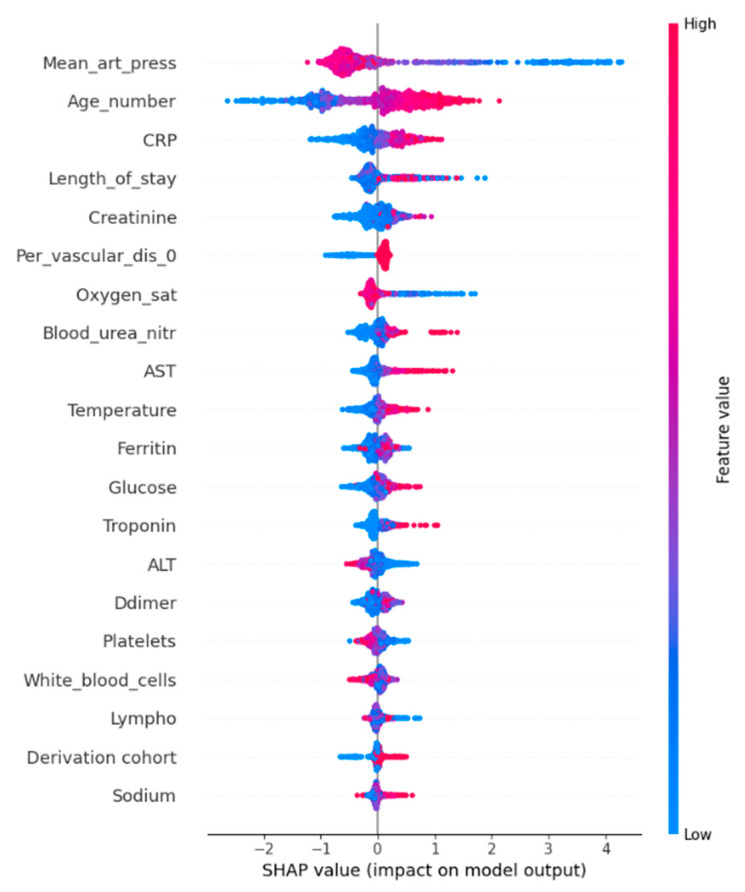
SHAP values for most important features for Mortality risk prediction.

**Figure 8 diagnostics-15-01981-f008:**
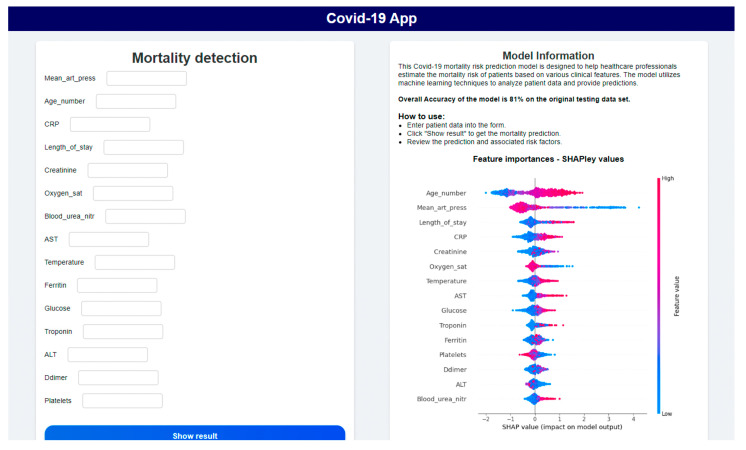
Section for Severity prediction in deployed application.

**Figure 9 diagnostics-15-01981-f009:**
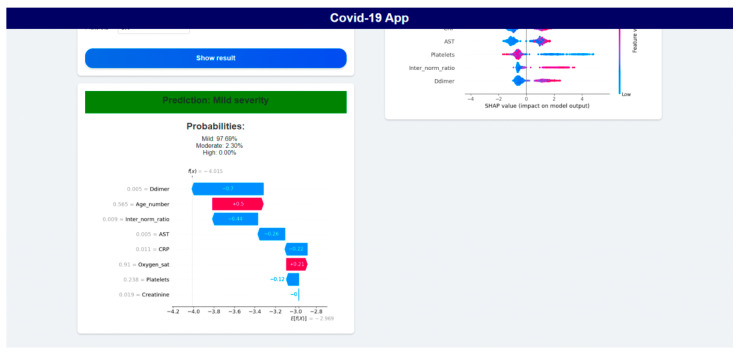
Severity prediction results and explanation in application.

**Table 1 diagnostics-15-01981-t001:** Description of features in dataset.

Feature	Description	Type
Derivation cohort	Dataset is divided into 2 cohorts—first 3 weeks of the pandemic and second 3 weeks of the pandemic	Binary
LOS	Length of stay = Duration of hospitalization in the hospital in days	Numerical
Death	Patient died during hospitalization	Binary
Severity	Severity of COVID-19 illness, expressed numerically in the range 0 to 11	Categorical
Black	Ethnicity	Binary
White	Ethnicity	Binary
Asian	Ethnicity	Binary
Latino	Ethnicity	Binary
Age.1	Age listed as a number	Numerical
AgeScore	Age category number (1, 2, 3, 4)	Categorical
MI	Myocardial infarction	Binary
PVD	Peripheral vascular disease	Categorical
CHF	Congestive heart failure	Binary
CVD	Cardiovascular disease	Binary
DEMENT	Dementia	Binary
COPD	Chronic obstructive pulmonary disease	Binary
DM Complicated	Diabetes mellitus complicated	Binary
DM Simple	Diabetes mellitus simple	Binary
Renal Disease	Kidney disease	Binary
All CNS	All diseases of the central nervous system	Binary
Pure CNS	Diseases only of the central nervous system	Binary
Stroke	Stroke	Binary
Seizure	Seizures	Binary
OldSyncope	Syncope (short-term loss of consciousness)	Binary
OldOtherNeuro	Other neurological diseases	Binary
OtherBrnLsn	Brain lesions	Binary
OsSats	Oxygen saturation = oxygen saturation in percentages	Numerical
Temp	Body temperature in degrees Celsius	Numerical
MAP	Mean arterial pressure, in mmHg	Numerical
Ddimer	Level of D-dimer in blood (mg/mL)	Numerical
Plts	Platelets = blood platelets, in thousands per mm^3^	Numerical
INR	International normalized ratio = laboratory value informing about the ability of blood to clot	Numerical
BUN	Blood urea nitrogen = blood urea nitrogen level, reflects kidney function (mg/dL)	Numerical
Creatinine	Creatinine level in blood. Reflects kidney function (mg/dL)	Numerical
Sodium	Sodium level in blood (mEq/L)	Numerical
Glucose	Blood glucose (mg/dL)	Numerical
AST	Aspartate aminotransferase in blood, in units/liter, liver tests	Numerical
ALT	Alanine aminotransferase in blood (units/L)	Numerical
WBC	White blood cells, number of white blood cells (in thousands per mm^3^)	Numerical
Lympho	Lymphocytes in blood (in thousands per mm^3^)	Numerical
IL6	Interleukin-6 in blood (in picogram/mL)	Numerical
Ferritin	Ferritin level in blood—a protein inside cells that stores iron. Indirectly measures iron level (microgram/L)	Numerical
CrctProtein	C-reactive protein level in blood (mg/L)	Numerical
Procalcitonin	Procalcitonin level in blood, ng/mL. Biomarker released in response to bacterial infections	Numerical
Troponin	Troponin level in blood, rises in myocardial infarction (ng/mL)	Numerical

**Table 2 diagnostics-15-01981-t002:** Chi-square test results of categorical independent features with *p*-value < 0.05 for Severity modeling.

Feature	*p*-Value	*p*-Value Interval
Age_category	0.0	<0.00001
CrtnScore	0.0	<0.00001
Death	1.71 × 10^−139^	<0.00001
PltsScore	4.72 × 10^−77^	<0.00001
Pure_CNS	3.87 × 10^−9^	<0.0001
All_CNS	6.03 × 10^−8^	<0.0001
Per_vascular_dis	0.003	<0.01
Derivation cohort	0.030	<0.05

**Table 3 diagnostics-15-01981-t003:** ANOVA results of numerical independent features with *p*-value < 0.05.

Feature	*p*-Value	*p*-Value Interval
Age_number	0.0	<0.00001
Blood_urea_nitr	0.0	<0.00001
Creatinine	3.21 × 10^−306^	<0.00001
Ddimer	2.61 × 10^−271^	<0.00001
CRP	2.15 × 10^−226^	<0.00001
Ferritin	1.29 × 10^−152^	<0.00001
Inter_norm_ratio	7.27 × 10^−150^	<0.00001
Mean_art_press	1.02 × 10^−134^	<0.00001
AST	2.27 × 10^−114^	<0.00001
Troponin	5.74 × 10^−76^	<0.00001
Oxygen_sat	4.46 × 10^−59^	<0.00001
Length_of_stay	2.0 × 10^−55^	<0.00001
Sodium	2.15 × 10^−45^	<0.00001
Lympho	4.78 × 10^−37^	<0.00001
Platelets	9.71 × 10^−32^	<0.00001
White_blood_cells	3.59 × 10^−30^	<0.00001
ALT	1.74 × 10^−18^	<0.00001
Glucose	1.07 × 10^−13^	<0.0001
Temperature	3.53 × 10^−8^	<0.001

**Table 4 diagnostics-15-01981-t004:** Results of models for prediction of severity.

Model	Random Forest	LightGBM	XGB	KNN	SVC	Logistic Regression	MLP
Accuracy	0.855	0.884	0.88	0.631	0.725	0.77	0.792

**Table 5 diagnostics-15-01981-t005:** Results of LightGBM model for Severity prediction trained on all features.

	Precision	Recall	F1-Score	Support
class 0	0.94	0.93	0.93	338
class 1	0.88	0.85	0.87	411
class 2	0.81	0.87	0.84	194
macro_avg	0.88	0.88	0.88	943
weighted_avg	0.89	0.88	0.88	943

**Table 6 diagnostics-15-01981-t006:** Results of simplified LightGBM model for severity prediction trained on 8 most important features.

	Precision	Recall	F1-Score	Support
class 0	0.95	0.94	0.94	339
class 1	0.88	0.88	0.88	400
class 2	0.84	0.86	0.85	204
macro_avg	0.89	0.90	0.89	943
weighted_avg	0.90	0,90	0.90	943

Class 0 represents low severity, class 1 medium severity, and class 2 high severity.

**Table 7 diagnostics-15-01981-t007:** Chi-square test results of categorical independent features with *p*-value < 0.05 for Mortality modeling.

Feature	*p*-Value	*p*-Value Interval
Age_category	2.17 × 10^−91^	<0.00001
CrtnScore	3.75 × 10^−67^	<0.00001
All_CNS	1.37 × 10^−9^	<0.0001
Per_vascular_dis	6.09 × 10^−9^	<0.0001
Pure_CNS	3.98 × 10^−8^	<0.0001
PltsScore	9.07 × 10^−6^	<0.0001
Stroke	2.22 × 10^−5^	<0.0001
Renal_dis	0.010	<0.01
Derivation cohort	0.023	<0.05
Chronic_obst_pulm_dis	0.027	<0.05
White	0.027	<0.05

**Table 8 diagnostics-15-01981-t008:** Chi-square test results of numerical independent features with *p*-value < 0.05.

Feature	*p*-Value	*p*-Value Interval
Mean_art_press	1.35 × 10^−171^	<0.00001
Blood_urea_nitr	1.15 × 10^−94^	<0.00001
CRP	8.99 × 10^−85^	<0.00001
Age_number	5.93 × 10^−66^	<0.00001
Ddimer	1.07 × 10^−60^	<0.00001
Creatinine	2.54 × 10^−47^	<0.00001
Ferritin	1.38 × 10^−37^	<0.00001
Oxygen_sat	2.38 × 10^−35^	<0.00001
Troponin	4.35 × 10^−31^	<0.00001
AST	7.33 × 10^−30^	<0.00001
Inter_norm_ratio	3.58 × 10^−25^	<0.00001
Glucose	1.73 × 10^−12^	<0.00001
Sodium	8.67 × 10^−12^	<0.00001
Lympho	2.15 × 10^−10^	<0.00001
White_blood_cells	2.80 × 10^−10^	<0.00001
Length_of_stay	3.87 × 10^−6^	<0.0001
Platelets	6.07 × 10^−6^	<0.0001
ALT	0.02	<0.05

**Table 9 diagnostics-15-01981-t009:** Models results for mortality prediction.

Model	Accuracy	F1-Score of Class 1	ROC AUC
Random Forest	0.8272	0.5329	0.837
LGBM	0.8244	0.5682	0.8316
XGB	0.8213	0.5655	0.8253
KNN	0.7519	0.3032	0.6792
Naïve Bayes	0.768	0.5009	0.7898
SVC	0.8007	0.407	0.7952
Logistic Regression	0.8191	0.5344	0.817
MLP	0.7642	0.4976	0.7607

**Table 10 diagnostics-15-01981-t010:** Performance of the best model for mortality risk prediction trained on all features.

Model	Precision	Recall	F1-Score	Support
Class 0	0.83	0.93	0.88	699
Class 1	0.71	0.46	0.56	244
Macro Avg	0.77	0.70	0.72	943
Weighted Avg	0.80	0.81	0.80	943

Note: Sensitivity (recall of Class 1) = 0.46; Specificity (recall of Class 0) = 0.93.

**Table 11 diagnostics-15-01981-t011:** Performance of the simplified model for mortality risk prediction trained on top 15 features.

Model	Precision	Recall	F1-Score	Support
Class 0	0.85	0.89	0.87	699
Class 1	0.64	0.57	0.60	244
Macro Avg	0.75	0.73	0.74	943
Weighted Avg	0.80	0.81	0.80	943

Note: Sensitivity (recall of Class 1) = 0.57; Specificity (recall of Class 0) = 0.89.

## Data Availability

The dataset used in this research for analysis and model training and testing is available at https://www.kaggle.com/datasets/harshwalia/mortality-risk-clinincal-data-of-covid19-patients (accessed on the 15 May 2024). The dataset is published under an open license for academic use, and all data are anonymized. The dataset from UNLP used for validation testing is not available for public sharing at this moment. The source code is available on GitHub at: https://github.com/takemenonemanikto/diagnostics_code (accessed on the 15 May 2024).

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
