# Peer review of "Data Analytics and Machine Learning Models on COVID-19 Medical Reports Enhanced with XAI for Usability"

_diagnostics, 2025, doi:10.3390/diagnostics15151981_

Round 1

Reviewer 1 Report

Comments and Suggestions for Authors

This study employs machine learning algorithms to forecast COVID-19 severity and death utilizing anonymized electronic medical records from 4,711 patients. The authors evaluate several models (LightGBM, Random Forest, XGBoost, etc.) and determine that LightGBM is the most effective model. They utilize SHAP values for model interpretability and implement streamlined models in a web application evaluated by medical professionals. The paper has potential but needs revisions to clarify its contribution, improve analysis, and address limitations.

Comments:

  1. The paper is relevant and addresses an important problem in healthcare, but its novelty is not very clear. Please explain how this work is different from existing studies using ML for COVID-19 prediction.

  2. The dataset comes from one hospital, and the external validation set (45 patients) is small. This limits generalizability. Please discuss this limitation more.

  3. The models have high accuracy but show low recall for mortality prediction (class 1). This imbalance issue should be better addressed with more metrics (sensitivity, specificity).

  4. The web app testing is a good idea, but there is little detail about the experts’ feedback. Were there any usability issues or suggestions for improvement?

  5. The paper is long in parts (especially “Related Work” and “Discussion”). Shorter and clearer text would improve readability.

Author Response

Thank you for the review. We have prepared a point-by-point response to all of your comments.

Comment 1:

Thank you for this important observation. We appreciate the opportunity to clarify the novel aspects of our study. While several previous studies have applied machine learning to COVID-19 data, our work contributes to the field in the following ways:

  1. Comprehensive Model Comparison: Unlike studies that focus on a single or limited set of algorithms, our work evaluates a broad range of models (Random Forest, LightGBM, XGBoost, CatBoost, SVM, KNN, Logistic Regression, and MLP), providing a more comprehensive comparative assessment.
  2. Model Simplification for Clinical Use: We go beyond theoretical modeling by developing simplified models using the most impactful features identified via SHAP and expert consultation. These simplified models maintain high performance while reducing input complexity, improving clinical applicability.
  3. Explainability with SHAP Integration: We place strong emphasis on model interpretability using SHAP values, not only during model analysis but also for individual prediction explanations in the deployed app, which enhances trust and transparency in clinical settings.
  4. Real-World Deployment and Expert Testing: Few studies have gone as far as deploying these models into a functioning web application that was tested with real hospital data and assessed by practicing clinicians. This practical deployment distinguishes our study from many that remain purely theoretical or retrospective. We have added a dedicated paragraph at the end of the Introduction section to better highlight these points of novelty. Please see the revised manuscript, starting at line 73.

Comment 2:

We appreciate the reviewer’s attention to the external validation component of our study. While the external validation cohort consisted of 45 patients, we emphasize that this validation was conducted using real-world, independently collected patient data from a different clinical setting and assessed in cooperation with practicing medical experts. This kind of practical, expert-tested validation adds a valuable dimension often missing in studies that focus solely on internal validation. Rather than being a limitation, we view this as a critical strength that supports the model’s applicability in practice. That said, we agree that expanding external validation in future work—particularly across multiple institutions and diverse populations—will further enhance robustness and broader applicability. We have clarified this perspective in the Discussion section of the manuscript, starting on line 646.

Comment 3:

We thank the reviewer for this valuable comment. Indeed, mortality prediction represents a class imbalance problem, as the number of deceased patients (class 1) is significantly lower than survivors (class 0). While we reported accuracy, F1-score for class 1, and ROC AUC, we agree that presenting sensitivity and specificity would provide a more complete understanding of the model's behavior—especially in a clinical context where false negatives are critical. We have now included sensitivity and specificity metrics for both the full and simplified LightGBM models used for mortality risk prediction. These metrics have been added under the Tables, according to model and discussed briefly in the revised text.

Comment 4:

We thank the reviewer for this excellent point. In our original manuscript, we briefly described the web application and its testing with domain experts, but we agree that further elaboration on the experts’ feedback—particularly regarding usability—is valuable. We have now expanded the relevant section to include qualitative feedback provided by the clinicians during and after testing. Although no major usability issues were reported, the experts did highlight specific strengths and minor suggestions for future development, such as improving the clarity of certain feature names and adding confidence intervals to predictions. These comments have been added to the revised manuscript.

Comment 5:

We appreciate the reviewer’s suggestion regarding clarity and conciseness. Although the Related Work and Discussion sections were already within standard length, we recognized that minor revisions could enhance readability. Accordingly, we have streamlined both sections by reducing redundancy and consolidating similar points while preserving essential details. For example, the Related Work section now groups similar studies and presents key findings more succinctly, and the Discussion has been tightened to focus on the main outcomes and their implications. We believe these revisions not only meet the reviewer’s suggestion for brevity and clarity but also maintain the technical rigor of the manuscript.

Reviewer 2 Report

Comments and Suggestions for Authors

The manuscript compares eight deep learning models in predicting the severity (three levels) and mortality (binary) of COVID-19 patients and later provides the best model as web apps that can be used by medical experts.

The dataset is from Harsh Walia from the Kaggle community, although the authors did not disclose the authority of the use of this dataset (and the data quality).

The study was weak in terms of the use of medical references/evidence and statistical reasoning, as many parts of the dataset only mention the opinion of medical experts, without qualitative or qualitative evidence

The study used chi-square and ANOVA to select the characteristics. However, later in the model development, all features were used to build the model and there were no results comparing model performance with different numbers of features (selected features and all features).

In the introduction, the rationale for using the features was different from that for developing the model, and the rationale was vague and weak (there was always mention of a medical expert, but there was no evidence). For example, line 195 (is there a medical reference to support this level of severity?); line 276 (what is the cut-off value/value for selecting the feature?); line 386 (what is the cut-off value for SHAP? Justification by a medical expert?).

In addition, the structure of the manuscript needs intensive revision to separate the methodology from the results, and the way the manuscript is presented is not up to standard. In addition, many methods were not clear. Where is the code for developing the models, how is LighGBM 'simplified', where is the link for the web application and how is the model used with Flask?

Comments on the Quality of English Language

The English could be improved to more clearly express the research.

Author Response

We thank the reviewer for the comments and valuable review of our manuscript. We prepared a point-by-point response to the comments. 

Comment 1:

We appreciate the reviewer’s attention to data transparency. However, we would like to clarify that the dataset used in this study was published publicly on Kaggle by Harsh Walia under the platform's default licensing terms, which allow for research, analysis, and publication use. The dataset is openly accessible and has been cited properly in the manuscript in Data availability section.

While the dataset is de-identified and secondary, its content is medically rich and has been used by multiple peer-reviewed studies. It includes structured clinical data collected from hospitalized COVID-19 patients, containing 85 features including lab measurements, comorbidities, and outcomes. Data quality was ensured through detailed preprocessing, including error detection (e.g., impossible values like negative temperatures), missing value handling using Multiple Imputation by Chained Equations (MICE), and expert consultation for identifying derived/supplementary variables.

We believe that the manuscript addresses the data authority or quality adequately. The manuscript explains the preprocessing methodology and explicitly discusses the removal of outliers and handling of missing values (Sections 3.1 and 3.2). Moreover, we emphasize that the source dataset is anonymized, ethically sharable, and suitable for machine learning applications focused on COVID-19 risk prediction.

To improve clarity, we have revised the Data Availability Statement (line 780) and added a brief note in Section 3 – Data Understanding (line 234) to explicitly reference the licensing and open-use status of the dataset.

Comment 2:

We thank the reviewer for raising this important point. In the revised manuscript, we have clarified that our approach integrates both statistical rigor and clinically grounded reasoning, with expert input used to complement—not replace—quantitative analysis.

Specifically:

  • Feature selection was performed using chi-square tests, ANOVA, t-tests, and LASSO regression. Expert review was used to confirm interpretability, not drive selection.
  • We now cite studies (e.g., Gao et al., 2020; Butler et al., 2020; Zucco et al., 2022) that support the clinical relevance of top features identified via SHAP values, such as oxygen saturation, D-dimer, and BUN.
  • Missing data was handled using a 30% feature exclusion threshold and Multiple Imputation by Chained Equations (MICE), as recommended by M.S. Khan (2022), F. Khan et al. (2022), and Fernández et al. (2025). Target variables were excluded from imputation to prevent leakage.
  • We now explain that SHAP values were used for feature ranking (line 393), not filtered by arbitrary thresholds. Selection for simplified models was based on a combination of consistent SHAP ranking and clinical interpretability, in line with explainable AI best practices (Salih et al., 2023).
  • Finally, we clarified in the Discussion (line 731) that expert guidance was used to contextualize findings and ensure practical relevance, while all modeling decisions were supported by data and literature.

Comment 3:

We thank the reviewer for the observation. We clarify that while the initial models were trained on the full feature set (53 features), we also developed simplified models using only the top SHAP-ranked features—8 for severity and 15 for mortality.

The comparative performance of these simplified models is shown in Table 10 and Table 11. In both cases, the simplified models achieved comparable or slightly better results than the full-feature models, demonstrating that fewer inputs can retain or improve performance while enhancing usability and interpretability.

These details are discussed in Sections 4.1.5 (line 392) and 4.2.6. We have revised the text to make this comparison more explicit and easier to locate (line 529).

Comment 4:

We appreciate the reviewer’s careful attention to methodological clarity.

  • Regarding severity classification (line 195): We now cite the WHO COVID-19 clinical guidance (2023), which supports the three-level severity structure used in our study.
  • Regarding feature selection (line 276): Features were selected using chi-square, ANOVA, t-tests, and LASSO, with further refinement based on SHAP value rankings and clinical interpretability. We did not apply hard cut-offs; rather, we used SHAP rankings as a guide, consistent with recommendations in explainable AI literature [Salih et al., 2023; Zucco et al., 2022].
  • Regarding SHAP thresholds (line 386): We clarified that SHAP values were not used with fixed numeric thresholds. Instead, we selected features with consistently high SHAP influence and clinical relevance, as also done in prior COVID-19 ML studies [Butler et al., 2020; Gao et al., 2020].

Comment 5:

We appreciate the reviewer’s suggestions regarding structure and transparency.

  • We have clarified the separation between the Methods and Results throughout the manuscript, particularly in Sections 3 and 4. Descriptions of modeling steps (e.g., SHAP, imputation, model simplification) were reframed to ensure methodological content is clearly distinguished from result reporting.
  • Regarding the LightGBM simplification, we now explicitly state that the simplified models use only the top-ranked features identified by SHAP analysis, selected based on both statistical influence and clinical interpretability (Sections 4.1.5 and 4.2.6).
  • A prototype web application was developed using the Flask framework in Python, demonstrating how the best-performing models can be deployed in practice. The application was tested internally by medical professionals to evaluate its usability. Due to institutional and privacy constraints, it is not publicly accessible, and no deployment link is provided.
  • Finally, we state in the manuscript that we plan to release a version of the source code on GitHub upon publication, including the core components for model development, evaluation, and deployment.

We believe these additions significantly improve the manuscript’s transparency, reproducibility, and overall presentation quality.

Reviewer 3 Report

Comments and Suggestions for Authors

To: Diagnostics MDPI

Dear EIC,

Dear AE,

I hope you are doing well.

This is my review report for the manuscript ID: diagnostics-3780481.

In this study, the authors evaluated several machine learning and data analytics models to analyze various types of medical reports related to patients with COVID-19. They aimed to predict the severity of COVID-19.

The main idea in the introduction is well-defined. The methods are scientifically described. The results are clearly presented and thoroughly discussed in the discussion section.

This is a valuable study that can add some novel things to the field.

Comments

  1. Please add the full name of the “ML” in the title.

Good luck

Author Response

We thank the reviewer for reviewing our manuscrip.

We also thank the reviewer for the helpful suggestion of updating the name. We have updated the title of the manuscript accordingly. The abbreviation “ML” has been replaced with its full form “Machine Learning.” The revised title now reads:

“Data Analytics and Machine Learning Models on COVID-19 Medical Reports Enhanced with XAI for Usability.”

Round 2

Reviewer 1 Report

Comments and Suggestions for Authors

The authors addressed all my concerns. The paper can be accepted for publication.

Reviewer 2 Report

Comments and Suggestions for Authors The authors have answered the question about the methodology (which features were used) and the validation of the model.

The manuscript has been significantly improved.